# Sn(IV) Sorption onto Illite and Boom Clay: Effect of Carbonate and Dissolved Organic Matter

Delphine Durce [1,*] , Sonia Salah [1], Liesbeth Van Laer [1], Lian Wang [1], Norbert Maes [1] and Stéphane Brassinnes [2]

1   Expert Group Waste & Disposal, Belgian Nuclear Research Centre (SCK-CEN), Boeretang, 2400 Mol, Belgium
2   ONDRAF-NIRAS, Belgian Agency for Radioactive Waste and Enriched Fissile Materials, Avenue des Arts 14, 1210 Bruxelles, Belgium
*   Correspondence: delphine.durce@sckcen.be; Tel.: +32-14-33-32-32

**Abstract:** [126]Sn is a long-lived fission product and it is important to assess its sorption onto the host rocks surrounding a possible nuclear waste repository. Boom Clay (BC) is under investigation in Belgium as a potential host rock. To better understand Sn(IV) sorption onto the clay minerals constituting BC, sorption of Sn(IV) was here investigated on Illite du Puy (IdP), from pH 3 to 12. Sorption isotherms at pH ~8.4 were acquired in the presence and absence of carbonate, and in the presence and absence of BC dissolved organic matter (DOM). Sn(IV) strongly sorbed on IdP over the full range of the pHs and concentrations investigated. In the presence of carbonates, Sn(IV) sorption was slightly decreased, highlighting the Sn(IV)–carbonate complexation. DOM reduced the Sn(IV) sorption, confirming the strong complexation of Sn(IV) with DOM. The results were modelled with the 2-site protolysis non-electrostatic surface complexation model. The surface complexation constants and aqueous complexation constants with carbonate and DOM were optimized to describe the experimental data. The applicability of the component additivity approach (CAA) was also tested to describe the experimental Sn(IV) sorption isotherm acquired on BC in BC pore water. The CAA did not allow accurate prediction of Sn(IV) sorption on BC, highlighting the high sensitivity of the model to the Sn(IV)-DOM complexation.

**Keywords:** tin; TSM; component additivity; ternary complexes; nuclear waste





## 1. Introduction

[126]Sn is a long-lived fission product and, as such, can represent a long-term radiological risk in the context of a nuclear waste repository. Due to its half-life of $2.3 \times 10^5$ years and its assumed low mobility through the repository barriers, it is considered to strongly contribute to the long-term activity inventory of vitrified and spent fuel wastes [1]. Within a safety case of a deep geological nuclear waste repository, it is, therefore, important to assess, as accurately as possible, the sorption of Sn onto the surrounding host rock, to ensure a good evaluation of its migration behaviour and radiological risk, while avoiding overconservative assumptions.

Unfortunately, both the hydro-geochemistry of Sn and its sorption onto the minerals constituting the host rocks and on the host rocks themselves, are poorly known. Sn(II) is considered to easily oxidize to Sn(IV) [2] and in the geochemical conditions expected in most of the nuclear waste repository scenarios, Sn(IV) would prevail [3,4]. However, due to the lack of thermodynamic data on Sn(IV) [2,5], the exact speciation of Sn(IV), under conditions relevant for geological disposal, remains highly uncertain, especially in the presence of inorganic and/or organic ligands. Due to the limited available thermodynamic data, scarce efforts have been devoted to assessing and modelling the sorption of Sn(IV) on pure clay minerals, such as illite and smectite. In clay-rich host rocks, such as those studied in France, Switzerland, Hungary and Belgium, 2:1 clay minerals are, in most cases, the main sorbing phases with respect to radionuclides [6]. Illite and montmorillonite are often taken as

representative for the sorption properties of the host rocks [6–8]. Bradbury and Baeyens [9] and Kedziorek et al. [4] investigated the Sn(IV) sorption onto montmorillonite and MX80 bentonite, respectively, while Bradbury and Baeyens [10] focused on the sorption on illite. All the studies reported sorption edges, i.e., as a function of pH (pH range from 2 to 12.5) and only Kedziorek et al. [4] investigated the effect of Sn(IV) concentration, but reported results over only a limited range from ~$4 \times 10^{-10}$ to ~$6 \times 10^{-9}$ mol/L. In all cases, Sn(IV) was found to strongly sorb on the studied clays, though a lower sorption was observed in the case of MX80 bentonite with log $R_d/K_d$ values at circumneutral pH (6 to 8.5) ranging from ~5.3 to ~5.6 for illite [10], from ~5.4 to ~6 for montmorillonite [9] and ~2 to ~4 L/kg for MX80 bentonite and Callovo-Oxfordian argillite [4]. Kedziorek et al. [4] did not observe a strong effect of pH on the sorption while Bradbury and Baeyens [9] and Bradbury and Baeyens [10] reported a decrease of sorption at pH > 8 and pH > 10 for montmorillonite and illite, respectively. The limited set of available experimental data, and their discrepancies, does not allow clearly assessment of the Sn(IV) sorption behaviour at different pHs, and even less over a range of Sn(IV) concentrations. Regarding the Sn(IV) sorption on clay-rich host rocks, to the best of our knowledge, the only available study is the one by Kedziorek et al. [4] on Callovo-Oxfordian argillite. With regards to the limited amount of data, it is clear that extra experimental data are needed to accurately assess Sn(IV) sorption onto clay-rich host rocks in repository conditions.

In Belgium, Boom Clay (BC) is under investigation as a potential host rock for deep geological disposal of nuclear waste. As with the other clay-rich host rocks, the main sorbing groups of minerals constituting BC are illite and smectite, though their content varies with depth [11]. The specificity of BC, in comparison with other investigated host rocks, is its relatively high concentration of organic matter (OM), which represents 1 to 5 wt%, and the high content of dissolved organic matter (DOM) found in the pore water at an average concentration of 108–115 mgC/L [12,13]. The strong binding of Sn(IV) with BC DOM was evidenced by Durce et al. [3], who derived Sn(IV)–DOM complexation constants from experiments performed in present-day BC conditions. Though the obtained constants were conditional to the investigated conditions, they were assumed to be representative of DOM found at present day in the Boom Clay pore water at the Mol site (Belgium), which is close to a 0.015 M $NaHCO_3$ solution at pH ~8.4, conditions referred to as "present-day BC conditions". Besides DOM, BC pore water at the Mol site also contains significant amounts of carbonates. The complexation of Sn(IV) to carbonate remains, to the best of our knowledge, unknown [5]. An analogy with Zr(IV), based on ionic radii, would suggest the formation of Zr–carbonate complexes, though in present-day BC conditions the hydrolysed species would still dominate the speciation [14].The presence of Sn(IV)-carbonate, even if expected to be limited, could play a role in the Sn(IV) speciation and, eventually, on the Sn(IV) sorption in present-day BC conditions.

To extend the knowledge on Sn(IV) sorption onto pure clay minerals and clay-rich host rocks, such as Boom Clay, the first objective of this work was to investigate, in batch sorption experiments, the Sn(IV) sorption onto Illite du Puy (IdP) originating from the clay-rich Puy-en-Velay Formation (Haute-Loire, France). The effect of various environmental factors, such as the presence of carbonate and DOM, both relevant for waste disposal into BC, was also studied.

The list of waste-relevant radionuclides is long and the sorption properties of many radionuclides on BC and on other clay-rich host rocks, could not be assessed because of technical, resource and/or time constraints. The so-called component additivity approach (CAA), or bottom-up approach, allows the use of data obtained on pure clay systems to be transferred to a range of conditions and host rock compositions, saving resources otherwise needed for studying each individual system. The CAA is a predictive modelling which combines, in an additive way, the sorption properties of the individual minerals constituting the natural system accounting for their relative concentrations [6,7,15]. Over the various models available in literature to describe the adsorption of metals and radionuclides on clay minerals, the quasi-mechanistic 2-site protolysis non-electrostatic surface complexation and

cation exchange model (2SPNE SC/CE) developed by Bradbury and Baeyens [15] is largely used in the context of nuclear waste disposal. For clay-rich host rocks, the CAA approach, based on the 2SPNE SC/CE model, was successfully applied to predict the sorption of several elements on the Hungarian Boda claystone [7,16], the French Callovo-Oxfordian (COx) claystone [8,17] and the Swiss Opalinus Clay [7,16]. In the case of BC, the CAA was only partially tested and the interaction of the RNs with BC OM was optimized in the experimental data [18].

The difficulty with the CAA is that as a sum of binary systems, it does not, in theory, integrate reactions specific to the ternary systems, such as the sorption of aqueous complexes. The presence of organic and/or inorganic ligands present in Boom Clay could, therefore, challenge the applicability of the CAA. Takahashi et al. [19], who investigated the sorption of several elements on kaolinite and silica in the presence of humic substances, observed that the coating of the solid surfaces by humic substances could alter the capacity of inorganic particles to adsorb metal. Few studies report the use of the CAA for ternary systems involving DOM. Bruggeman et al. [20] were able to successfully describe the Eu(III) sorption on illite in the presence of BC DOM using the CAA including the Tipping model [21]. On the other hand, Kaplan et al. [22] faced more difficulties in representing the sorption of Eu(III) on sediments. The authors, indeed, observed at low DOM concentrations an increase of Eu(III) sorption in comparison to the predictions of the CAA. They attributed this discrepancy to a possible increase of the sediment sorption affinity for Eu(III) due to sorbed DOM and/or to a modification, with contact with the sediment, of the DOM affinity towards Eu(III). Similar observations were reported by Warwick et al. [23] and Evans et al. [24].

In present-day Boom Clay conditions, Sn(IV) speciation is expected to be mainly controlled by the presence of DOM, though a complexation with the carbonates present in the pore water cannot be excluded. In such conditions, the validity of the CAA to predict Sn(IV) sorption onto BC is uncertain. The second objective of this study was to assess the sorption of Sn(IV) onto Boom Clay in present-day BC conditions and to test the applicability of the CAA. The tested CAA used the 2SPNE SC model calibrated on the sorption results obtained on Illite du Puy (IdP). The aqueous Sn(IV) complexation with BC DOM and carbonate were either predicted from literature data, if available, and/or from the sorption experiments performed on IdP in the presence of BC DOM and carbonates.

## 2. Materials and Methods

All the experimental uncertainties are given as $2\sigma$ or 2 standard deviation (SD) (95% confidence intervals).

### 2.1. Chemicals and Reagents

All solutions were prepared with ultra-pure deionized water (18.2 M$\Omega$.cm) and with commercially available chemical products of analytical grade or superior.

The reference for "inert" conditions was 0.017 M NaClO$_4$. The effect of carbonate on the Sn(IV) sorption was evaluated in 0.015 M NaHCO$_3$. NaClO$_4$ and NaHCO$_3$ solutions were prepared from the corresponding salts, i.e., NaClO$_4$ anhydrous (Acros Organics, Geel, Belgium) and NaHCO$_3$. H$_2$O (Merck, Darmstadt, Germany), dissolved in degassed ultrapure water directly in an Ar-atmosphere glovebox ($\leq$2 ppm O$_2$). Stock solutions of higher concentration, i.e., 0.023 M for NaClO$_4$ and 0.02 M for NaHCO$_3$, were made to be further diluted when preparing the clay suspensions and the samples for the sorption experiments. To avoid the disturbance of the illite, KCl salt was also added to the stock solutions to a concentration of $1.3 \times 10^{-4}$ mol/L. The effect of DOM on the Sn(IV) sorption was investigated in 0.017 M NaClO$_4$ in the presence of $4.5 \pm 2.8$ ppmC of BC DOM.

BC DOM was extracted from the pore water collected from the EG/BS piezometer installed in the HADES underground laboratory (Mol, Belgium). The extraction procedure was similar as the one described in Durce et al. [3]. The extracted DOM was recovered in a small volume of 0.017 M NaClO$_4$ and further purified by dialysis at 1 kDa against 0.017 M

NaClO$_4$ (Regenerated cellulose membrane Spectra/Por7, 18∗11.5 mm). The dialyzed solution was then stored in a dark bottle in the glovebox until further use and referred to as DOM_NaClO$_4$. The concentration of dissolved organic carbon in DOM_NaClO$_4$ was determined by TOC measurements performed on a TOC-L (Shimadzu) carbon analyser, as described in Durce et al. [3]. The size distribution of DOM_NaClO$_4$ was measured by size exclusion chromatography, as described in Durce et al. [13]. The obtained size distribution is reported in the Supplementary Materials (Figure S1). The concentration of natural Sn in DOM_NaClO$_4$ was not quantified, but DOM_NaClO$_4$ was assumed to have similar properties as the batches isolated in Durce et al. [3]. The Sn content was, therefore, assumed to be <1 × 10$^{-6}$ mol/gC. The nature and concentration of cationic impurities and the proton exchange capacity (PEC) were also assumed equal to the values of the BC DOM, referred to as Batch2_NaHCO3_02 in Durce et al. [3], with an effective PEC value of 10.7 ± 2.1 meq/gC.

The Sn(IV) sorption on BC was investigated in BC pore water. The pore water was collected from the four filters of the horizontally-oriented piezometer TD-116E, installed at the level of the HADES laboratory (−196.6 m Tweede Algemene Waterpassing). The water collected was given the official reference TD-116E-all/2021-05-19, but, for the sake of simplicity, in this paper it is referred to as SPRING water. The content of organic and inorganic carbon (TOC and TIC) was measured, as described in Durce et al. [3]. The measured values were 133 ± 27 mg/L and 82 ± 25 mg/L for TIC and TOC, respectively. The ionic composition of the SPRING water is reported in Table S2 of the Supplementary Materials. Prior to the sorption experiments, the SPRING water was filtered at 0.45 μm (PVDF hydrophilic, VWR).

To allow working at trace Sn(IV) concentrations, a radioactive source of $^{113}$Sn (half-life of 115.09 days) was used. A source of SnCl$_4$ in 6 M HCl was purchased from Eckert and Ziegler with a specific activity of 7.5 × 10$^5$ Bq/mL (at the date of purchase) and a carrier concentration of 4.21 × 10$^{-4}$ mol/L. Two $^{113}$Sn solutions were purchased and used over the study duration. The total concentration of Sn in the first $^{113}$Sn solution was not crosschecked and was assumed to match the value given by the supplier. The second solution was measured by ICP-MS and displayed a Sn concentration of (3.01 ± 0. 30) × 10$^{-4}$ mol/L.

Various sub-solutions of the original spikes were prepared by dilution in ultrapure water or HCl and used in the experiments. For practical reasons, the solutions were prepared outside the glovebox, but were always allowed to equilibrate afterwards with the atmosphere of the glovebox (Ar).$^{113}$Sn is a quasi-mono-energetic gamma emitter in the energy region around 390 keV and was analysed by gamma-counting on a Packard Cobra Quantum counter in the energy range 300–500 keV. The counting efficiency ($\varepsilon$) was measured by Durce et al. [3] to 0.35 ± 0.02 and revalidated here. Later on in the study, it was observed to drop to 0.29 ± 0.02, probably due to a lower specific activity of the second purchased $^{113}$Sn source. The total concentration (mol/L) of Sn in the samples was calculated from the measured activity as:

$$[Sn] = \frac{(CPM_s - CPM_b)}{V_s \times 60 \times \varepsilon \times e^{\left(-\frac{ln2}{t_{1/2}} \times \Delta t\right)}} \times \frac{C_0}{A_0} \tag{1}$$

with $CPM_s$ and $CPM_b$, the measured counts per minute of the sample and the blank, respectively, $V_s$ (mL), the measured volume of sample, $t_{1/2}$ (days), the half-life of $^{113}$Sn (days), $\Delta t$ (days), the time span between the day of measurement and the reference date on the activity certificate, $C_0$ (mol/L), the Sn concentration of the source and $A_0$ (Bq/mL), the initial activity of the $^{113}$Sn source. The uncertainty associated to the Sn(IV) concentration was calculated by propagating the experimental errors (confidence limit of 95%) with $2\sigma(\frac{C_0}{A_0})/\frac{C_0}{A_0} = 0.1$, corresponding to the measurement error on $C_0$, $2\sigma(V_s)/V_s$ which was neglected and assumed null, $2\sigma(\varepsilon)/\varepsilon = 0.06$ corresponding to the average of the experimental error on $\varepsilon$.

### 2.2. Illite and Boom Clay Samples

The illite used in this work was the Illite du Puy from the clay-rich Puy-en-Velay Formation (Haute-Loire, France). The batch used here was prepared within the CatClay EU project [25] during which it was purified using a multistep process and saturated in sodium (Na) using the method of Baeyens and Bradbury [26] and Bradbury and Baeyens [10]. The obtained high purity Na-form IdP sieved at 63 μm was used in the following experiments. Various characterisations of the purified IdP were performed within the framework of CatClay, and for more information we refer the reader to Altmann et al. (2015). Only the BET specific surface area was measured in the present work by $N_2$-adsorption measurements and was found to be equal to $115 \pm 12 \ m^2/g$.

The Boom Clay (BC) sample used in the sorption experiments originated from the non-oxidized BC core referred to as CG72–73W_core 13_Section 13.1 and was sampled within the Putte Member region (12.40–13.35 m intrados) of Boom Clay. The TIC and TOC contents of the BC core were measured at RTW university (Aachen, Germany) and corresponded to $0.11 \pm 0.03$ and $1.23 \pm 0.03$ mg/kg, respectively. The mineralogical composition (Table 1) was determined by quantitative XRD (X-ray diffraction) analysis performed at KUL university (Leuven, Belgium) with the use of ZnO as internal standard and the full pattern summation method for data quantification. The BET specific surface area measured by $N_2$-adsorption was found to be equal to $47 \pm 5 \ m^2/g$.

**Table 1.** Composition of the BC core CG72–73W_core 13_Section 13.1 measured by XRD on two subsamples.

| Minerals | Concentration (wt% $\pm$ Spread) [1] |
|:---:|:---:|
| Plagioclase | $1.5 \pm 0.5$ |
| Calcite | $0.7 \pm 0.2$ |
| Pyrite | $1.5 \pm 0.5$ |
| Anatase | $0.9 \pm 0.1$ |
| K-feldspar | $6 \pm 1$ |
| Quartz | $34.5 \pm 7.5$ |
| **SUM Non-Clay** | $\mathbf{45 \pm 9}$ |
| Kaolinite | $8 \pm 2$ |
| 2:1 Al Clay | $35 \pm 4$ |
| Chlorite | $1.5 \pm 1.5$ |
| Muscovite | $10.5 \pm 1.5$ |
| **SUM Clay** | $\mathbf{55 \pm 9}$ |

[1] The values correspond to the average and spread of the two measurements calculated as spread $= |\text{average} - \text{measurement}|$.

Two stock suspensions of IdP < 63 μm, referred to as IdP_0 and IdP_1 and one of 'peeled' BC, referred to as BC_1, were prepared in an Ar-glovebox at a solid/liquid ratio of ~20 g/L in 0.017 M $NaClO_4$ + $1 \times 10^{-4}$ mol/L KCl for the IdP suspensions and SPRING water was used for the BC suspension. The suspensions were stirred for at least 1 day and the equilibrium solutions were analysed by ICP-MS by the Division of Soil and Water Management of KUL university (Leuven, Belgium). The cationic composition of the solutions is reported in the Supplementary Materials (Table S1). The measured concentration of natural Sn in the suspensions was $(6.01 \pm 1) \times 10^{-9}$ and $(2.05 \pm 0.2) \times 10^{-8}$ mol/L in IdP_0 and BC_1, respectively. It was below detection limit in IdP_1. The concentration measured in IdP_0 was also close to detection limit and outside the calibration range. Considering the relatively low value and its large uncertainty, the concentration of natural Sn in the IdP sorption experiments was assumed negligible. In the case of BC suspensions, natural Sn was more concentrated and could be present both on the BC solid as well as in the SPRING water. Consequently, as described further on, the concentration of natural Sn present in blank suspensions prepared in parallel to the sorption experiment samples, was also measured.

Both IdP and BC suspensions contained a significative amount of metals that could compete with Sn(IV) for sorption (Table S1). However, Sn(IV) has a different chemistry and hydrolysis behaviour than the metals contained in the suspensions (mainly divalent) and the competition between them was expected to be rather limited. Studies on competitive sorption onto clays indeed showed that competition tended to be selective and to occur only for elements with similar chemistry [27–29].

### 2.3. Sorption Edge on Illite

A sorption edge of Sn(IV) on IdP was acquired at room temperature (22 °C) over 18 pH points ranging from pH 3 to pH 12 in 0.017 M $NaClO_4$, at a Sn(IV) concentration of $(6.8 \pm 0.4) \times 10^{-8}$ mol/L and a solid/liquid ratio of $0.65 \pm 0.02$ g/L. The pH was maintained with the use of acid acetic, MES (2-(*N*-Morpholino)ethane-sulphonic acid), MOPS (3-(*N*-Morpholino) propanesulphonic Acid), TRIS (Tris(hydroxymethyl)aminomethane) and CHES (3-(cyclohexyl amino)ethanesulphonic acid) buffers. Bradbury and Baeyens [10] reported the absence of effect of the buffers on Th(IV) sorption on IdP. It was assumed that the buffers also did not affect Sn(IV) sorption. The concentration of the buffers in the experiments was fixed at $2 \times 10^{-3}$ mol/L and stock solutions of 0.2 mol/L were prepared in ultrapure water. For pH < 4.5 and >10, no pH buffer was added to the samples and the pH was adjusted to the target value with the addition of NaOH or $HClO_4$ 0.5 M.

In an Ar-glovebox, the right volumes of $NaClO_4$ solution, pH buffer, base or acid (NaOH 0.5 M or $HClO_4$ 0.5 M) and $^{113}$Sn solution were added to the tubes to a total volume of 19.5 mL. The tubes were then shaken on a mechanical shaker for 1 h and the pH was controlled and adjusted if needed. Then, 0.5 mL of the vigorously stirred IdP_0 suspension was added and the tubes were shaken again for 1 h. The pH was controlled once more and adjusted if needed. The tubes were put under agitation on a mechanical shaker for 7 days, which, according to the results of Bradbury and Baeyens [10], was sufficient to reach sorption equilibrium. An amount of 1 mL of the suspensions was then sampled and analysed by gamma counting. The suspensions were homogenized by vigorous shaking before or during sampling. The tubes were taken out of the glovebox, centrifuged for 1 h at $20,000\times g$ and gently reintroduced into the glovebox for sampling. Assuming an average density of illite particles of 2.75 [30], the centrifugation step should settle particles down to 20 nm. The pH of the supernatants was measured and the supernatants analysed by gamma counting. The Sn(IV) sorption on the tubes and the possible Sn(IV) loss during centrifugation was assessed by running three blank solutions in parallel with respective pH of 3, 7 and 12. The blanks were prepared following the same procedure as the samples, but without addition of the clay suspension.

### 2.4. Sorption Isotherms on Illite

Sorption isotherms of Sn(IV) on IdP were acquired at room temperature (22 °C) in 0.017 M $NaClO_4$ in the presence and absence of $4.5 \pm 2.8$ ppmC of DOM_$NaClO_4$ and in 0.015 M $NaHCO_3$. The sorption of Sn(IV) was measured for the different conditions over seven concentration points with three replicates per concentration. Two solid/liquid ratios (~0.1 and ~0.5 g/L) were used to cover the range of Sn(IV) concentrations.

The protocol followed for the sorption isotherms was the same as described for the sorption edge with the use of TRIS buffer to adjust the pH to ~8.4 and $NaClO_4$ or $NaHCO_3$ solutions. For the isotherm in the presence of DOM_$NaClO_4$, 0.1 mL of DOM_$NaClO_4$ solution was added to the tubes after the addition of $^{113}$Sn. Aliquots of ~0.1 or 0.5 mL of IdP_1 suspension were added and the total volume of solution was ~20 mL. The range of investigated Sn(IV) concentrations, the used solid/liquid ratios and the averaged measured equilibrium pHs are reported in Table 2. For all investigated conditions, three blank solutions were prepared in parallel at an average Sn(IV) concentration of $(2.9 \pm 0.2) \times 10^{-8}$ mol/L. The blanks were prepared following the same procedure as the samples, but without addition of the clay suspension.

**Table 2.** Experimental conditions for Sn(IV) sorption isotherms on illite (IdP). Sn(IV) concentration corresponds to the concentration introduced at the start of the experiments.

| Conditions | Sn(IV) Concentration Range (mol/L) | Solid/Liquid Ratio (g/L) ($\pm$2 SD) | Average pH Measured at 7 Days ($\pm$2 SD) |
|---|---|---|---|
| 0.017 M NaClO$_4$ | $3.3 \times 10^{-9}$ to $4.2 \times 10^{-7}$ | $0.10 \pm 0.002$, $0.51 \pm 0.02$ and $0.53 \pm 0.02$ | $8.56 \pm 0.1$ and $8.49 \pm 0.04$ |
| 0.017 M NaClO$_4$ + 4.5 ppmC DOM_NaClO$_4$ | $2.0 \times 10^{-9}$ to $3.0 \times 10^{-7}$ | $0.10 \pm 0.02$ and $0.46 \pm 0.04$ | $8.37 \pm 0.06$ |
| 0.015 M NaHCO$_3$ | $1.7 \times 10^{-9}$ to $3.0 \times 10^{-7}$ | $0.09 \pm 0.02$ and $0.52 \pm 0.02$ | $8.66 \pm 0.32$ |

Due to a significant loss of Sn(IV) by what we assumed to be side sorption on the tubes, the sorption isotherm obtained in NaClO$_4$ covered a narrower concentration range than the ones obtained in the presence of ligands. To solve the issue, extra points at higher Sn(IV) concentrations were performed at a later stage. The average solid/liquid ratio and equilibrium pHs of these extra points are reported in Table 2. The contact time was reduced to 4 days, which was judged, from a limited time-dependent study, to be sufficient to reach the sorption equilibrium.

*2.5. Sorption Isotherm on Boom Clay*

The sorption isotherm of Sn(IV) on BC was acquired in SPRING water with Sn(IV) concentrations ranging from $4.8 \times 10^{-9}$ to $6.2 \times 10^{-7}$ mol/L. Only one solid/liquid ratio was used with an average value of $0.13 \pm 0.01$ g/L for a total volume solution of ~20 mL. The experimental protocol was the same as reported for the sorption isotherms on IdP except for the centrifugation step, which was, in the case of BC, done for 2 h at $20{,}000\times g$. The average equilibrium pH was measured as $8.41 \pm 0.04$. As for the other experimental sets, three blank solutions were run in parallel with an average Sn(IV) concentration of $(2.5 \pm 0.2) \times 10^{-8}$ mol/L. In addition, two blank suspensions were also run in parallel to estimate the concentration in solution of natural Sn. The blank suspensions were prepared following the same procedure as the samples, but without addition of $^{113}$Sn. At the end of the contact time, the blank suspensions were centrifuged for 2 h at $20{,}000\times g$ and the supernatant analysed for Sn concentration by ICP-MS by the Division of Soil and Water Management of KUL university. The average natural Sn concentration measured in the equilibrium solution was $(5.50 \pm 0.24) \times 10^{-9}$ mol/L.

More information on the weighed volumes used in each sample and the blanks can be found in the Supplementary Materials.

**3. Modelling**

The extent of sorption of Sn(IV) is here represented by the solid–liquid distribution coefficient referred to as $R_d$ (L/kg), and calculated according to Equation (2):

$$R_d = \frac{Sn_0 - Sn_{eq}}{Sn_{eq}} \times \frac{V}{m} \tag{2}$$

with $Sn_0$ (mol/L), the concentration in the suspension before centrifugation, $Sn_{eq}$ (mol/L), the concentration in the supernatant after centrifugation, $V$ (L), the volume of solution and $m$ (kg), the mass of clay. Both concentrations were calculated based on the measured activities using Equation (1).

The calculated $R_d$ values were provided with their associated uncertainties calculated by propagating the experimental errors (confidence limit of 95%) as described in the Supplementary Materials.

### 3.1. Modelling of Sn(IV) Sorption on Illite

The thermodynamic sorption model used to describe the obtained experimental data was implemented in the PHREEQC Version 3 geochemical code [31] and the calculations were done with the Thermochimie V10a database [32] using the extended Debye-Huckel equation (B-dot).

The experimental sorption data were modelled using the 2SPNE SC/CE model developed by Bradbury and Baeyens (1997) to describe the sorption of cations on Montmorillonite SWy and further extended to IdP (Bradbury and Baeyens, 2009). The protolysis constants determined by Bradbury and Baeyens [10] were used here and are reported in Table 3. The uncertainty on the protolysis constants was taken as 0.5 log unit. Bradbury and Baeyens [10] did not perform a formal uncertainty estimation. However, the protolysis constants being determined on an iterative modelling approach, including sorption data, we assumed here that the largest uncertainty on these data, usually taken as 0.5 log unit [9], propagated to the protolysis constants. The Sn(IV) hydrolysis constants implemented in Thermochimie V10a database and derived from the review of Gamsjäger et al. [5] were used in all the modelling exercises (Table 4). In agreement with these constants, Sn(IV) was expected to start hydrolysing from low pH and the contribution of cation exchange to its sorption could be assumed negligible [9]. Sn(IV) cation exchange was, therefore, not implemented in the model and only surface complexation was considered. In the range of concentrations investigated here, Sn(IV) was assumed to sorb only on the strong-type sites of IdP and, though the weak sites were implemented in the model, they did not contribute to Sn(IV) sorption. The capacity of strong sorption sites was allowed to vary and was adjusted on the experimental data (Table 3). The surface complexation reactions taken up in the model were determined in agreement with the Sn(IV) speciation. They are reported in Table 5.

**Table 3.** Protolysis reactions and constants and surface site capacities for Illite du Puy.

| Protolysis | Bradbury and Baeyens [10] [1] | This Work [2] |
|---|---|---|
| $\equiv S^{w1}OH + H^+ \leftrightarrow \equiv S^{w1}OH_2^+$ | $log\ K = 4.0 \pm 0.5$ | |
| $\equiv S^{w2}OH + H^+ \leftrightarrow \equiv S^{w2}OH_2^+$ | $log\ K = 8.5 \pm 0.5$ | |
| $\equiv S^s OH + H^+ \leftrightarrow \equiv S^s OH^{2+}$ | $log\ K = 4.0 \pm 0.5$ | |
| $\equiv S^{w1}OH \leftrightarrow \equiv S^{w1}O^- + H^+$ | $log\ K = -6.2 \pm 0.5$ | |
| $\equiv S^{w2}OH \leftrightarrow \equiv S^{w2}O^- + H^+$ | $log\ K = -10.5 \pm 0.5$ | |
| $\equiv S^s OH \leftrightarrow \equiv S^s O^- + H^+$ | $log\ K = -6.2 \pm 0.5$ | |
| **Surface site capacity** | | |
| $\equiv S^{w1}OH$ | $4 \times 10^{-2}$ mol/kg | |
| $\equiv S^{w2}OH$ | $4 \times 10^{-2}$ mol/kg | |
| $\equiv S^s OH$ | $2 \times 10^{-3}$ mol/kg | $1.0\ [0.97–1.1] \times 10^{-3}$ mol/kg |

[1] Derived at 25 °C and I = 0 M, [2] optimized from experimental data at 25 °C and I = 0.017 M.

**Table 4.** Sn(IV) hydrolysis reactions and constants implemented in Thermochimie V10a database and derived from Gamsjäger et al. [5] compared to constant values used by Bradbury and Baeyens [10].

| Sn(IV) Hydrolysis | In Thermochimie V10a [1] | In Bradbury and Baeyens [10] [1] |
|---|---|---|
| $Sn^{4+} + H_2O \leftrightarrow SnOH^{3+} + H^+$ | / | $log\ K^1_{OH} = 1.2$ |
| $Sn^{4+} + 2H_2O \leftrightarrow Sn(OH)_2^{2+} + 2H^+$ | / | $log\ K^2_{OH} = 1.7$ |
| $Sn^{4+} + 3H_2O \leftrightarrow Sn(OH)_3^+ + 3H^+$ | / | $log\ K^3_{OH} = 1.6$ |
| $Sn^{4+} + 4H_2O \leftrightarrow Sn(OH)_4 + 4H^+$ | $log\ K^4_{OH} = 7.54 \pm 0.69$ | $log\ K^4_{OH} = 0.4$ |
| $Sn^{4+} + 5H_2O \leftrightarrow Sn(OH)_5^- + 5H^+$ | $log\ K^5_{OH} = -1.06 \pm 0.80$ | $log\ K^5_{OH} = -7.7$ |
| $Sn^{4+} + 6H_2O \leftrightarrow Sn(OH)_6^{2-} + 6H^+$ | $log\ K^6_{OH} = -11.13 \pm 0.76$ | $log\ K^6_{OH} = -18.4$ |

[1] Derived at 22 °C and I = 0.017 Mc.

**Table 5.** Sn(IV) surface complexation and aqueous complexations reactions and their corresponding constants implemented in the sorption model.

| Sn(IV) Surface Complexation and Aqueous Complexation | | Literature Data | This Work [3] |
|---|---|---|---|
| $\equiv S^s OH + Sn^{4+} + 4H_2O \leftrightarrow \equiv S^s OSn(OH)_4{}^- + 5H^+$ | $\log K_3$ | 2.5 [1] | 9.4 [8.8–10.1] |
| $\equiv S^s OH + Sn^{4+} + 5H_2O \leftrightarrow \equiv S^s OSn(OH)_5{}^{2-} + 6H^+$ | $\log K_4$ | $-5.7$ [1] | 0.9 [0.7–1.4] |
| $Sn^{4+} + 4CO_3^{2-} \leftrightarrow Sn(CO_3)_4{}^{4-}$ | $\log K_5$ | / | 55.8 [55.7–56.1] |
| $Sn^{4+} + DOM\_s^- - \leftrightarrow SnDOM\_s^{3+}$ | $\log K_s^{**(Sn-DOM)}$ | 50.0 [2] | / |
| $Sn^{4+} + DOM\_w^- - \leftrightarrow SnDOM\_w^{3+}$ | $\log K_w^{**(Sn-DOM)}$ | 46.0 [2] | / |
| $Sn^{4+} + DOM^- - \leftrightarrow SnDOM^{3+}$ | $\log K_6$ | / | 45.4 [45.2–46.0] |
| $Sn^{4+} + DOM\_s^- + \equiv S^s OH - \leftrightarrow \equiv S^s OSnDOM\_s^{2+} + H^+$ | $\log K_7$ | / | 48.9 [48.5–49.4] |
| $Sn^{4+} + DOM\_w^- + \equiv S^s OH - \leftrightarrow \equiv S^s OSnDOM\_w^{2+} + H^+$ | $\log K_8$ | / | 48.9 [48.5–49.4] |

[1] Data available from Bradbury and Baeyens [10] but with a different set of Sn(IV) hydrolysis data; [2] recalculated from Durce et al. [3] using Equations (5) and (6); [3] optimized constants with the estimated confidence interval.

Information on Sn(IV)–carbonate complexation is scarce and no thermodynamic constant was validated in the review of Gamsjäger et al. [5]. The absence of a thermodynamic constant describing this complexation makes the prediction of the effect of carbonate on the Sn(IV) aqueous speciation difficult. Sn(IV) has an ionic radius most similar to Zr(IV) and would be expected to follow Zr(IV) geochemical behaviour. Despite some gaps remaining in the knowledge of Zr(IV) complexation with carbonate, the formation of $Zr(CO_3)_4{}^{4-}$ and the corresponding constant were experimentally validated [33]. By analogy, it was chosen to write the Sn(IV)–carbonate complexation with the formation of $Sn(CO_3)_4{}^{4-}$ as reported in Table 5.

Durce et al. [3] measured the Sn(IV) complexation with BC DOM in present-day BC conditions, i.e., in 0.015 M NaHCO$_3$ and at a pH of $8.34 \pm 0.08$. The complexation data were modelled using a 2-sites non-linear Langmuir isotherm by considering the following simple reactions:

$$Sn_{inorg} + DOM\_s \leftrightarrow SnDOM\_s \qquad logK_s^{(Sn-DOM)} = 8.80 \tag{3}$$

$$Sn_{inorg} + DOM\_w \leftrightarrow SnDOM\_w \qquad logK_w^{(Sn-DOM)} = 4.85 \tag{4}$$

where $DOM\_s$ and $DOM\_w$ would represent strong and weak binding sites present on BC DOM, respectively, and with a ratio of $DOM\_s/DOM\_w = (1.06 \pm 0.31) \times 10^{-3}$.

The experimental conditions of Durce et al. [3] and of the present work are close with respect to pH and ionic strength, which allowed the use of the conditional complexation constants reported by the authors. The Sn(IV)–DOM complexation reactions were here implemented in the sorption model with $Sn^{4+}$ as main species, according to the formalism reported in Table 5. The complexation constants were recalculated from Durce et al. [3], taking into account the hydrolysis behaviour of Sn(IV) (Table 4) and the complexation with carbonate (Table 5), according to Equations (5) and (6):

$$LogK_{s/w}^{**\,(Sn-DOM)} = LogK_{s/w}^{(Sn-DOM)} + LogA_s \tag{5}$$

$$A_s = 1 + \frac{K_{OH}^1}{[H^+]} + \frac{K_{OH}^2}{[H^+]^2} + \frac{K_{OH}^3}{[H^+]^3} + \frac{K_{OH}^4}{[H^+]^4} + \frac{K_{OH}^5}{[H^+]^5} + \frac{K_{OH}^6}{[H^+]^6} + \left[CO_3^{2-}\right]^4 \times K_5 \tag{6}$$

With $A_s$, the side reaction coefficient and $\left[CO_3^{2-}\right]$, the concentration of carbonate at the pH of Durce et al. [3] experiments estimated at $1.50 \times 10^{-4}$ mol/L.

The model was then adapted to better describe the experimental sorption of Sn(IV) in the presence of DOM_NaClO$_4$. Two options were considered. One option was to implement the Langmuir isotherm reported by Durce et al. [3] with the recalculated complexation constants as reported in Table 5, but integrating also the sorption of the formed Sn(IV)-DOM

complexes to the model. Simple 1:1 reactions were considered as reported in Table 5. The surface complexation constants for the two Sn(IV)-DOM complexes, *SnDOM_s* and *SnDOM_w*, were referred to as $K_7$ and $K_8$ and it was assumed that $K_7 = K_8$. A second option was a simplified ligand model accounting for only one binding site on BC DOM. Sn(IV) complexation with BC DOM was, in that case, implemented with the reaction reported in Table 5 and the associated constant, *log* $K_6$.

The calculated *log* $R_d$ values were modelled by adjusting the relevant constants and the sorption site capacity. The sorption isotherm and sorption edge obtained in 0.017 M NaClO$_4$ were fitted simultaneously by optimizing the Sn(IV) surface complexation constants, *log* $K_3$ and *log* $K_4$ and the IdP strong site capacity. The Sn(IV)–carbonate complexation constant, *log* $K_5$ and the Sn(IV)–DOM complexation constant, *log* $K_6$ or Sn(IV)-DOM surface complexation constants, *log* $K_7$ and *log* $K_8$, were optimized on the experimental sorption data obtained in 0.015 M NaHCO$_3$ and in 0.017 M NaClO$_4$ + DOM_NaClO$_4$, respectively. The optimisation was done using the UCODE_2014 program [34] with or without the use of singular-value decomposition and by minimizing the weighted least-squares objective function with respect to the constant values, using a modified Gauss–Newton method. The uncertainty on the determined constants and the sorption site capacity was estimated from a sensitivity analysis performed as described in the Supplementary Materials (Figures S4–S9). The boundary of the confidence interval for each fitted parameter was taken as the largest negative and positive variations induced by the uncertainty on the selected input parameters.

### 3.2. Component Additivity Approach for Sn(IV) Sorption on Boom Clay

The optimised constants were used to describe the Sn(IV) sorption on Boom Clay following the component additivity approach. Sn(IV) sorption is not affected by cation exchange and the competitive sorption with the metals present in SPRING water was assumed negligible, as previously discussed. The formation of Sn(IV) sulfato-complex was also neglected with respect to the low sulfate content of SPRING water (Table S2). For simplicity, the composition of SPRING water was, therefore, approximated to a NaHCO$_3$ solution of 0.015 M. To evaluate the contribution of each mechanism on the sorption of Sn(IV), the model was built in a stepwise manner, gradually increasing its complexity. In a first step, only the contribution of the surface complexation on the 2:1 clay mineral was integrated. The average concentration of 2:1 clay mineral in the sample used in this study was measured by XRD analysis to 35 wt% (Table 1). This value accounted for illite concentration and illite/smectite mixed layer concentration. Marques Fernandes et al. [7] showed that approximating the sorption behaviour of the illite/smectite layer to the one of pure illite led to a relatively good representation of the sorption of several elements onto two argillaceous rocks. Dähn et al. [16] confirmed this result for Zn sorption at low sorbate concentration. In agreement with these two studies, it was assumed here that Sn(IV) sorption onto the illite/smectite mixed layer present in BC had the same characteristics as on pure illite. The surface site concentrations of BC were, therefore, assumed to correspond to 35% of the IdP site capacity. In a second step, the model was adapted to integrate the Sn(IV)–carbonate complexation. In a last step, Sn(IV)–DOM complexation and the potential formation of DOM–Sn(IV)-BC complexes were added to the model. Two models were tested. In a first exercise, only the aqueous Sn(IV)–DOM complexation was integrated in the CAA with the optimized *log* $K_6$ constant value. In a second exercise, the aqueous Sn(IV)–DOM complexation was described with the constants $logK_s^{**(Sn-DOM)}$ and $logK_w^{**(Sn-DOM)}$ and the sorption of the formed complexes was added to the model with the optimized constants *log* $K_7$ and *log* K$_8$. The SPRING water contained $82 \pm 25$ ppm of DOC, which translated to $(8.77 \pm 3.18) \times 10^{-4}$ mol/L, considering the value of $PEC_{eff}$. DOM in SPRING water differs from DOM_NaClO$_4$ as it contains a large fraction of small molecules (<1 kDa [13]) which have been filtered out during the extraction of DOM_NaClO$_4$. It was, however, considered for this modelling exercise that DOM_NaClO$_4$ and the DOM present in SPRING water displayed the same complexation properties with respect to Sn(IV).

## 4. Results

### 4.1. Sorption of Sn(IV) on Illite in 0.017 M NaClO₄

The experimental sorption edge of Sn(IV) on IdP in 0.017 M NaClO$_4$ is reported in Figure 1. The sorption of Sn(IV) on IdP was very strong over the full pH range. It was, indeed, in the same order of magnitude as the sorption of other tetravalent elements, such as Th(IV) and U(IV) [17,35]. However, the experimental sorption edge for Sn(IV) did not show a strong 'edge', which was in agreement with the hydrolysis occurring already at low pH. The decrease of sorption at pH < 4 was in line with the assumed negligible contribution of cationic exchange. On the other side of the pH range, the clear decrease of sorption at pH > 10 resulted from the formation of Sn(OH)$_6^{2-}$, which does not strongly sorb on IdP.

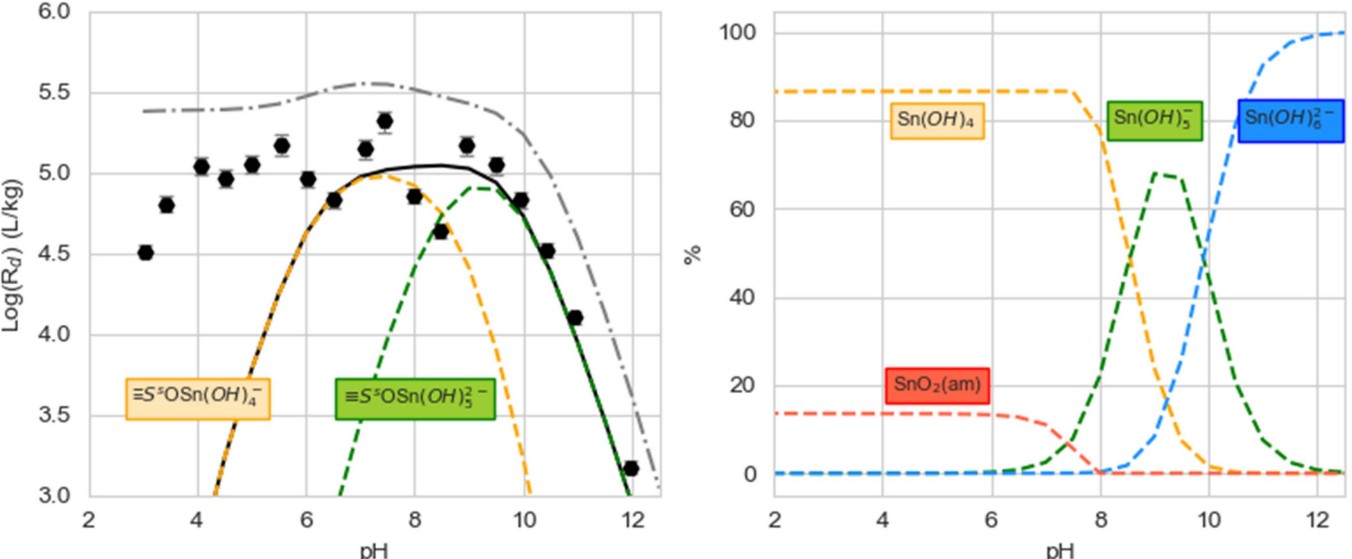

**Figure 1. Left**: Sorption edge of Sn(IV) on IdP in 0.017 M NaClO$_4$ at initial Sn(IV) concentration of (6.8 ± 0.4) × 10$^{-8}$ mol/L and solid/liquid ratio of 0.65 ± 0.02 g/L. Black markers = experimental points. Grey dashed line = 2SPNE SC model with Sn(IV) surface complexation constants and hydrolysis constants reported by Bradbury and Baeyens [10] (Table 4). Black solid line = 2SPNE SC model with *log* $K_3$, *log* $K_4$ (Table 5) and ≡ $S^sOH$ (Table 3) and hydrolysis constants from Thermochimie V10a (Table 4). Dotted coloured lines = contribution of the different surface complexes. **Right**: Sn(IV) speciation at 6.8 × 10$^{-8}$ mol/L in 0.017 M NaClO$_4$ calculated with Thermochimie V10a.

To describe their data Bradbury and Baeyens [10] used a different set of Sn(IV) hydrolysis constants than used here (Table 4). They used data reported by Baes and Mesmer (1977) that they converted to log K using the Davies relation (1962) combined with Sn$^{4+}$ values published by Hummel et al. (2002). The resulting model using these set of hydrolysis constants and the Sn(IV) surface complexation constants that they determined is reported in Figure 1. It overestimates the experimental *log* $R_d$ values of up to 1 log unit.

The experiments of Bradbury and Baeyens [10] were performed at a higher ionic strength, i.e., 0.1 M. However, in the absence of cation exchange, ionic strength is not expected to strongly affect Sn(IV) sorption. It would also lead to an opposite trend to the one observed here, i.e., higher sorption at lower ionic strength. On the one hand, the lower sorption observed here could be due to different properties of the used IdP batch. The sorption site capacities and hydrolysis constants of the batch used in the present study were not measured, but the reported cation exchange capacity (CEC), 190 ± 10 meq/g [25] was smaller than the one of the IdP batch of Bradbury and Baeyens (2009), which was measured to 225 ± 15 meq/g. The difference in CEC, though limited, indicated the existence of some differences between the two IdP batches. Some of these differences could possibly have impacted the Sn(IV) sorption.

The experimental protocol followed by Bradbury and Baeyens [10] is not reported in details, but the observed different Sn(IV) uptake between the two studies could also result from the use of different experimental protocols. As shown in Table A1 (Appendix A), the loss of Sn(IV) measured in the absence of IdP in 0.017 M NaClO$_4$ showed a strong Sn(IV) sorption on the tube walls at all pH-values (up to 91%), while the loss, during the centrifugation step at pH > 7, was minimal (<6%). By measuring the activity of $^{113}$Sn of the suspensions just before centrifugation, the bias due to Sn(IV) sorption on tube walls was here limited. However, a strong loss of Sn(IV) during centrifugation at pH < 7 was observed in the blank solutions. This could also have occurred in the presence of IdP and biased the calculated *log $R_d$* values. One should therefore keep in mind that the sorption on IdP could be overestimated at pH < 7. Though not expected, the different centrifugation conditions applied here and in Bradbury and Baeyens [10] could also impact the final results. Bradbury and Baeyens [10] used a higher centrifugation speed than here, which could, on the one hand, ensure a stronger solid/liquid separation, but, on the other hand, could settle potential Sn(IV) colloids. The lower Sn(IV) sorption here observed on IdP in comparison to Bradbury and Baeyens [10] could, therefore, have been due to a combination of differences in the used material and the used experimental approach.

Despite the difference of magnitude, the Sn(IV) sorption on IdP followed here the same trend for almost the entire pH range as reported by Bradbury and Baeyens [10]. It was, however, different from Kedziorek et al. [4], who did not observe any pH effect on the sorption of Sn(IV) on MX80 bentonite. The best description of our experimental sorption edge and isotherm was obtained with the Sn(IV) surface complexation constants reported in Table 5 and the IdP strong site capacity reported in Table 3. The predominance of Sn(OH)$_4$ from low pH as predicted using the hydrolysis constants implemented in the Thermochimie V10a database (Figure 1), would result in a sorption decrease for pH < 7 concomitant with the protonation of the IdP sorption sites. The apparent high sorption of Sn(IV) observed down to pH 2 could indicate the involvement of lower hydrolysis species, as suggested by the modelling put forward by Bradbury and Baeyens [10]. However, due to the high uncertainty on the Sn(IV) hydrolysis data used by the authors and to the lack of validated thermodynamic data on Sn(IV) behaviour in acidic conditions still persisting today, it is difficult to conclude on the role of Sn(OH)$^{3+}$, Sn(OH)$_2$$^{2+}$ and Sn(OH)$_3$$^+$. On the other hand, the strong loss of Sn(IV) with centrifugation, observed in the blank solutions at pH < 7, indicated a possible precipitation. As shown in Figure 1 (right), at pH < 8, SnO$_2$(am) started to precipitate with a maximum of 13.6% of Sn(IV) precipitated at pH 2. The extent of precipitation was limited and could even be lower in the presence of IdP, but it could partially explain the apparent high sorption observed at low pH.

Although, the implemented surface complexation model was not able to properly describe the experimental data obtained at low pH, it satisfactorily depicted the sorption of Sn(IV) for pH > 7. These pH conditions are the most relevant for Boom Clay and, more generally, for a nuclear waste repository, and the sorption model was, therefore, applicable to this context.

As visible on the experimental sorption isotherm in 0.017 M NaClO$_4$, presented in Figure 2, sorption was linear until an equilibrium Sn(IV) concentration of $1.6 \times 10^{-9}$ mol/L, over which the distribution coefficient ($R_d$) started to decrease, indicating a possible site saturation. The sole optimisation of the Sn(IV) surface complexation constants failed in reproducing the saturation effect (Figure 2, black dotted line), and to properly describe the experimental results (Figure 2, black line), the capacity of the strong sorption sites of IdP had to be reduced by a factor two in comparison to the values reported by Bradbury and Baeyens [10] (Table 3). Such a reduction of the site capacity was not reported in the modelling exercises performed in the context of Catclay [25], for which the same IdP batch was used. However, only divalent and trivalent elements were investigated in the frame of the project. The strong hydrolysis of Sn(IV) could prevent it gaining access to the totality of the IdP strong sorption sites. Tetravalent elements usually display a low solubility at near-neutral pH and above and sorption studies in these pH conditions are

usually limited to low concentrations, to avoid precipitation [7,17,35–37]. In these low concentration ranges, site saturation is not expected and it is difficult to compare the here observed Sn(IV) behaviour with other reported data. At the high Sn(IV) concentration points investigated, $SnO_2$,am was expected to precipitate, as shown in the Eh-pH diagram reported in the Supplementary Materials (Figures S2 and S3). However, in the presence of IdP, the Sn(IV) equilibrium concentration dropped below the solubility limit of $SnO_2$,am and precipitation was, therefore, excluded.

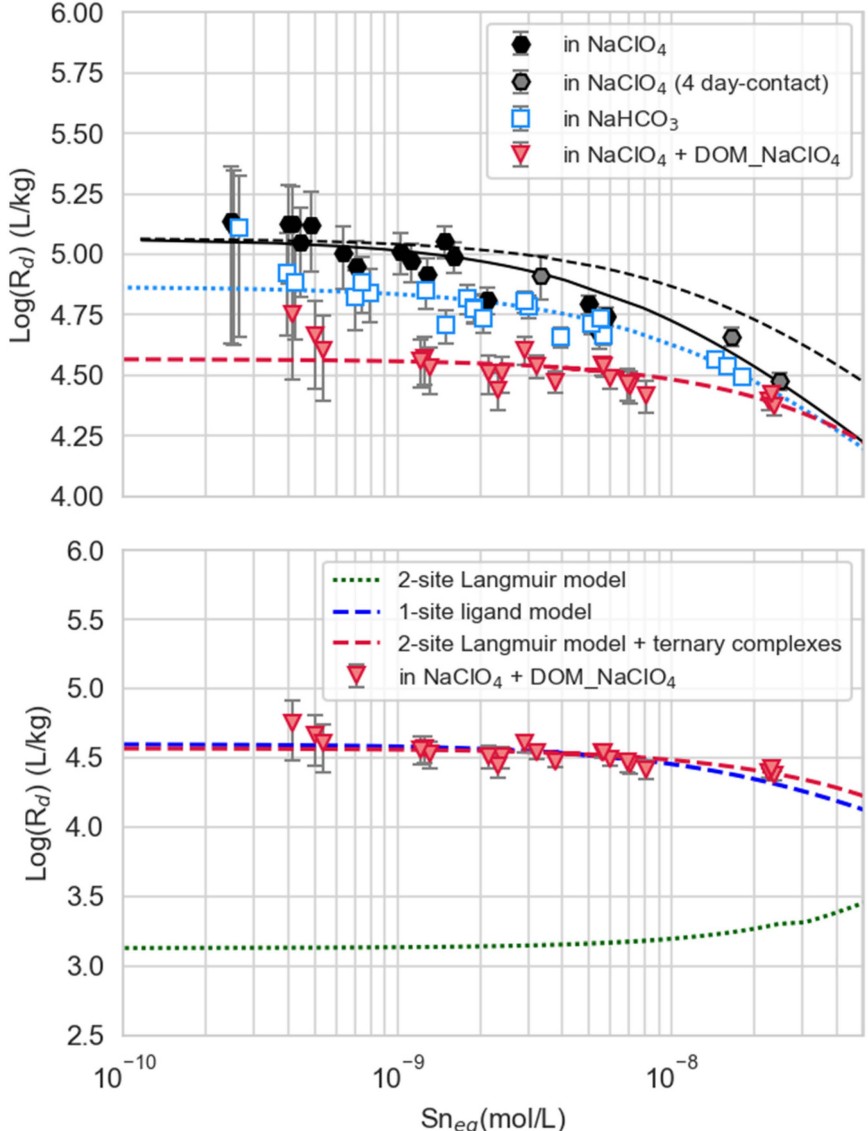

**Figure 2. Up:** Sorption isotherms of Sn(IV) on IdP in 0.017 M $NaClO_4$, 0.015 M $NaHCO_3$ and 0.017 M $NaClO_4$ + 4.5 ppmC DOM_$NaClO_4$. Markers = experimental points, black dotted line = 2SPNE SC model (*log* $K_3$ = 9.1 and *log* $K_4$ = 0.6), black solid line = 2SPNE SC model (*log* $K_3$, *log* $K_4$ (Table 5), $\equiv S^sOH$ (Table 3)), blue dotted line = 2SPNE SC model (*log* $K_3$, *log* $K_4$ (Table 5), $\equiv S^sOH$ (Table 3)) + Sn(IV)-carbonate complexation (*log* $K_5$), red dashed line = 2SPNE SC model (*log* $K_3$, *log* $K_4$ Table 5), $\equiv S^sOH$ (Table 3)) + Sn(IV)-DOM complexation (*log* $K_s^{**(Sn-DOM)}$, *log* $K_w^{**(Sn-DOM)}$ (Table 5)) + Clay-Sn(IV)-DOM complexation (*log* $K_{7/8}$ (Table 5)). **Bottom:** Sorption isotherms of Sn(IV) on IdP in 0.017 M $NaClO_4$ + 4.5 ppmC DOM_$NaClO_4$. Green dotted line = 2SPNE SC model (*log* $K_3$, *log* $K_4$ (Table 5), $\equiv S^sOH$ (Table 3)) + + Sn(IV)-DOM complexation (*log* $K_s^{**(Sn-DOM)}$, *log* $K_w^{**(Sn-DOM)}$ (Table 5)), dark blue dashed line = 2SPNE SC model (*log* $K_3$, *log* $K_4$ (Table 5), $\equiv S^sOH$ (Table 3)) + Sn(IV)-DOM complexation (*log* $K_6$ (Table 5)).

### 4.2. Sorption of Sn(IV) on Illite in 0.015 M NaHCO₃

The sorption isotherm of Sn(IV) on IdP in 0.015 M NaHCO$_3$ is reported in Figure 2. In comparison with the NaClO$_4$ system, the sorption of Sn(IV) on IdP was slightly reduced in the presence of carbonates. Although the resulting decrease of the *log R$_d$* values remained small, and did not exceed 0.3 log unit, it was consistent over the full range of investigated concentrations. Assuming that the surface complexation constants and the site capacity optimised on the NaClO$_4$ system applied here, the best fit to the experimental data was obtained by taking into account the carbonate–Sn(IV) complexation with the constant value as reported in Table 5. The nature and the stoichiometry of the Sn(IV)–carbonate complexation reaction(s) occurring in our system was unknown and more data would be needed to determine it. The formation of $Sn(CO_3)_4^{4-}$ together with the value of *log K$_5$* should be considered more as fit parameters than physical parameters. The obtained constant was indeed highly conditional on the experimental conditions used here and to the formalism chosen for the complexation reaction. Nevertheless, if one considers the sorption decrease as significant, the Sn(IV) complexed to the carbonates would represent, in the investigated conditions, 39% of the total aqueous Sn(IV). This was significantly more than for Zr(IV), and though hydrolysis still dominated the Sn(IV) speciation, the complexation with carbonate should not be neglected. More experiments to assess both the binary Sn(IV)-carbonate and the ternary Sn(IV)-carbonate-clay systems would be necessary to get a clear picture on the effect of carbonate on the Sn(IV) sorption.

### 4.3. Sorption of Sn(IV) on Illite in 0.017 M NaClO₄ in Presence of Dissolved Organic Matter

The sorption isotherm of Sn(IV) on IdP in 0.017 M NaClO$_4$ in the presence of DOM_NaClO$_4$ is presented in Figure 2. In comparison with the pure NaClO$_4$ system, the presence of BC DOM reduced the sorption of Sn(IV) on IdP over most of the range of investigated concentrations with a decrease in *log R$_d$* of up to 0.5 log unit.

The strong complexation of Sn(IV) with BC DOM was evidenced in Durce et al. [3] and was shown to depend on the loading, i.e., the DOM/Sn(IV) ratio. A two-site Langmuir isotherm type model was used to fit the experimental data. The corresponding constants reported by Durce et al. [3] were recalculated, taking into account the Sn(IV) hydrolysis constants and carbonate complexation reported in Tables 4 and 5, respectively. The integration of the Langmuir isotherm and the recalculated constants in the sorption model significantly overestimated the sorption decrease induced by the presence of BC DOM (Figure 2, bottom). The model also predicted an increase in *log R$_d$* values at the highest Sn(IV) concentrations, due to the saturation of the complexation site of high affinity on BC DOM. However, this increase was not visible in the experimental data.

The apparent lower effect of BC DOM on the Sn(IV) sorption on IdP than predicted using the data reported by Durce et al. [3], could be due to a bias introduced by the recalculation of the Sn(IV)-DOM complexation constant, or to a poor representability of these constants of the present system or to the formation of ternary complexes Clay-Sn(IV)-DOM.

Durce et al. [3] experimentally measured the amount of Sn(IV) complexed to DOM in solution by a combination of ultrafiltration and gamma counting at different DOM concentrations. The complexation constants were then calculated based on this amount and on the total non-complexed Sn(IV), without making any assumption on the speciation of the non-complexed Sn(IV). The recalculation of these constants to the formalism, as described in the Modelling section, was, therefore, highly dependent on the used Sn(IV) hydrolysis constants and on the Sn(IV)–carbonate complexation constant. Yet, because the experiments of Durce et al. [3], and our experiments were performed at similar pH conditions, the bias introduced by the constant recalculation was expected to be limited.

The BC DOM used by Durce et al. [3] was extracted following the same protocol as here, but it was redissolved in 0.015 M NaHCO$_3$ instead of 0.017 M NaClO$_4$. The *PEC$_{eff}$* of the batch used in the present work was not measured and was assumed equal to the *PEC$_{eff}$* measured by Durce et al. [3]. The possible differences in the two batches could affect the properties of BC DOM with respect to Sn(IV) complexation and render the complexation

constant reported by Durce et al. [3] not applicable to the present system. Allowing the DOM–Sn(IV) complexation constant to vary, the best fit of the sorption data was obtained by considering one complexation site on BC DOM with the complexation constant as reported in Table 5. The optimized value of $log\,K_6$ was 0.6 log unit lower than $log\,K_w{}^{**(Sn\text{-}DOM)}$. The lower constant, together with the absence of a strong complexation site, would indicate a significantly lower affinity of DOM_NaClO$_4$ towards Sn(IV) than the DOM batch used by Durce et al. [3].

　　　The formation of ternary complexes Sn(IV)-DOM-Clay or DOM-Sn(IV)-Clay could also explain the apparent discrepancy with the results of Durce et al. [3]. The stoichiometry of the DOM–Sn(IV) complexes is not elucidated and the complexes could sorb on IdP with Sn(IV) acting as a cation bridge between the deprotonated IdP and BC DOM sites. The sorption of BC DOM itself was reported by Bruggeman et al. [20] to be low and the coverage of the IdP surface by BC DOM should therefore be limited. DOM_NaClO$_4$ was added to the suspension (quickly) after Sn(IV) and in such conditions, we expected that the Sn(IV) complexation with DOM in solution would dominate over the complexation to DOM sorbed on the clay surface. Indeed, Takahashi et al. [19] showed that the order of addition had a strong influence on the distribution of Zr and Hf on kaolinite in the presence of humic acid. The formation of DOM–Sn(IV)-Clay ternary complexes mediated by Sn(IV) was tested by implementing in the model simple 1:1 reactions with similar constants for the formation of $\equiv S^sOSnDOM\_s^{2+}$ and $\equiv S^sOSnDOM\_w^{2+}$. The best fit to the experimental data was obtained with the constant value as reported in Table 5. The sorption of Sn(IV) was fully controlled by the sorption of the $SnDOM\_w^{3+}$ complexes, while the model was insensitive to the formation of $\equiv S^sOSnDOM\_s^{2+}$.

　　　The modelled data described well the experimental points, both by modifying the complexation constant DOM-Sn(IV) and by introducing the formation of ternary complexes DOM-Sn(IV)-Clay. It was, however, rather unexpected that two batches of DOM originating from the same pore water would behave so differently with respect to their affinity for Sn(IV). Humic and fulvic acids showed, in general, consistent complexation behaviour independent of their origin. Generic parameters have indeed been derived from semi-mechanistic complexation models and successfully applied in several studies [37–42]. The difference observed with the prediction based on the results of Durce et al. [3] would, therefore, indicate the existence of additive mechanisms specific to the ternary system Sn(IV)/Clay/ BC DOM. The formation of ternary complexes DOM-Sn(IV)-clay would be one of these mechanisms. This result was in contradiction with the conclusion of Bruggeman et al. [20], who neglected the contribution of BC DOM sorption and ternary complex formation in their model to describe the sorption of Eu(III) on illite. However, tetravalent elements are expected to behave differently than trivalents and Sn(IV) to display different sorption and complexation behaviour than Eu(III). On that basis, we here assumed that the formation of ternary complexes DOM-Sn(IV)-Clay, controlled, at least partly, the Sn(IV) sorption on IdP in the presence of DOM. The forming mechanisms of these complexes and their stoichiometry remain, however, unknown and the reactions reported in Table 5 are hypothetical. Yet, they allowed taking into account the sorption of Sn(IV)–DOM complexes and to further evaluate its contribution on Sn(IV) sorption onto BC. Deeper investigations would be required to confirm the presence of these ternary complexes and unravel their exact nature.

### 4.4. Sorption of Sn(IV) on Boom Clay in Clay Water and Component Additivity Approach

　　　The sorption isotherm of Sn(IV) on BC in SPRING water is reported in Figure 3. The experimental data are presented either as a function of $Sn_{eq}$, as calculated from the measured activity (Equation (1)) or recalculated to take into account the concentration of Sn natural, as measured in the blank suspensions. The presence of natural Sn affects only the low concentration points, where it shifts the sorption isotherm towards higher equilibrium Sn concentrations. Except for the obvious outlier, the sorption of Sn(IV) on BC was linear until an equilibrium concentration of ~$2 \times 10^{-8}$ mol/L. Beyond, the $log\,R_d$ values started

to gradually decrease, possibly due to a gradual saturation of the strong sorption sites. In comparison with the pure clay system, the Sn(IV) sorption on BC was lower, which could be expected, considering that only 35 wt% of BC was made of 2:1 clay minerals. Yet, as shown in Figure 3, the sorption was overestimated when only taking into account in the model the BC 2:1 clay mineral content and the optimised Sn(IV) sorption properties of IdP (Table 5). The experimental results also showed that site saturation shifted to higher Sn(IV) concentration than predicted by the model. Sorption of Sn(IV) on a second type of sorption sites of lower affinity but larger capacity could become significant at high Sn(IV) concentrations. The concentration range investigated on IdP was, however, too low to allow the determination of Sn(IV) surface complexation constants on this type of site.

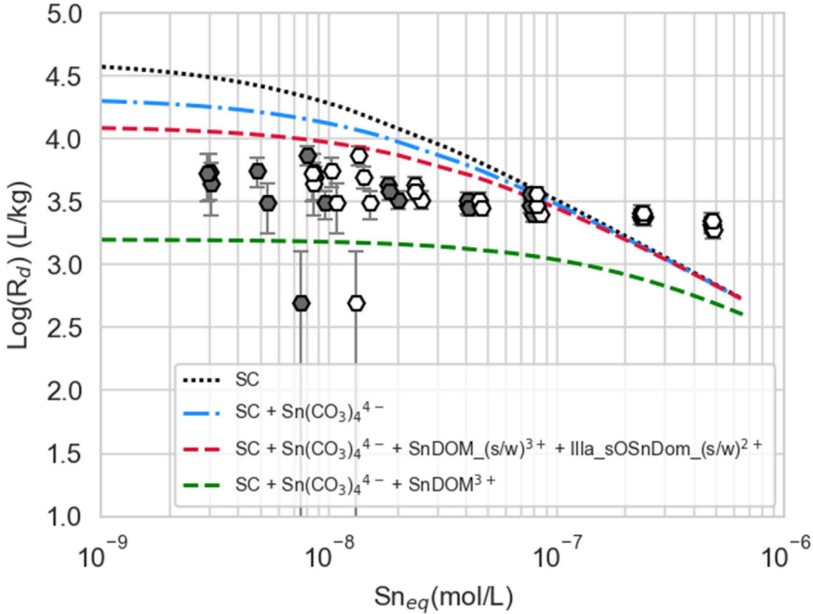

**Figure 3.** Sorption isotherm of Sn(IV) on Boom Clay in SPRING clay water at pH= $8.41 \pm 0.04$ and at a solid/liquid ratio of $0.13 \pm 0.01$ g/L. Grey markers= experimental data, empty markers= experimental data recalculated taking into account the natural Sn concentration such as $Sn_{eq}(recalculated) = Sn_{eq} + Sn_{natural}$. Black dotted line = 2SPNE SC model (*log* $K_3$, *log* $K_4$ (Table 5), $\equiv S^sOH$ (Table 3)), blue dashed line = 2SPNE SC model (*log* $K_3$, *log* $K_4$ (Table 5), $\equiv S^sOH$ (Table 3)) + Sn(IV)-carbonate complexation (*log* $K_5$, Table 5), green dashed line = 2SPNE SC model (*log* $K_3$, *log* $K_4$ (Table 5), $\equiv S^sOH$ (Table 3)) + Sn(IV)-carbonate complexation (*log* $K_5$, Table 5) + Sn(IV)-DOM complexation (*log* $K_6$, Table 5), red dashed line = 2SPNE SC model (*log* $K_3$, *log* $K_4$ (Table 5), $\equiv S^sOH$ (Table 3)) + Sn(IV)-carbonate complexation (*log* $K_5$, Table 5) + Sn(IV)-DOM complexation (*log* $K_s^{**(Sn-DOM)}$, *log* $K_w^{**(Sn-DOM)}$ (Table 5)) + Clay-Sn(IV)-DOM complexation *log* $K_{7/8}$ (Table 5).

Adding the Sn(IV)–carbonate complexation with the *log* $K_5$ value determined on IdP (Table 5) improved the fit, but the Sn(IV) uptake was still overestimated in comparison to the experimental results. The integration in the model of Sn(IV) complexation with BC DOM using the value of *log* $K_6$ optimised from the IdP experiments (Table 5), predicted a strong reduction of the *log* $R_d$ values and led to an underestimation of the experimental data. On the other hand, integrating the formation of DOM-Sn(IV)-ternary complexes as determined on IdP resulted in an overestimation of the Sn(IV) sorption. In present-day BC conditions, the speciation of Sn(IV) is mostly controlled by its complexation to DOM and so is its sorption onto clay minerals. The model was highly sensitive to the presence of Sn(IV)–DOM complexes and to their potential sorption on the clay surface. None of the modelling approaches calibrated on the IdP system, i.e., Sn(IV) surface complexation with DOM-Sn(IV) aqueous complexation with or without sorption of the formed com-

plexes, was able to accurately predict the Sn(IV) sorption on BC. SPRING DOM mainly contains medium to small organic species (<20 kDa, Durce et al. [13]) and is different to DOM_NaClO$_4$ which contains larger molecules. SPRING DOM could display a different reactivity towards Sn(IV) than DOM_NaClO$_4$, leading to an apparently lower DOM contribution in the Sn(IV) sorption onto BC. On the other hand, contrary to the IdP system, BC DOM was in equilibrium with BC and the presence of DOM coated to the BC surface and/or of solid organic matter could affect the extent and the nature of the ternary Sn(IV)-DOM-Clay complexes.

Due to the strong complexation of Sn(IV) to DOM in the investigated conditions, the heterogeneity of BC DOM and the uncertainty on the aqueous DOM chemistry and on the formation of Sn(IV)-DOM-Clay ternary complexes, the CAA as used here appeared not robust enough to accurately predict Sn(IV) sorption onto BC. However, it already gave access to a rough estimation of the Sn(IV) sorption with a prediction of the log $R_d$ values which differed only by a maximum of 0.5 log unit from the experimental values.

## 5. Discussion

It has long been known that, with its high concentration of NOM and especially DOM, BC is a special case amongst the investigated clay host rocks. The migration of several RNs was shown to be controlled by their complexation to BC DOM [43]. As part of the safety case, the RNs are grouped according to their chemical properties and the group "RNs with a transport behaviour linked to DOM" contains a large number of RNs, including Sn [44]. In the presence of rather simple and well-known aqueous chemistry, the use of the CAA provided fairly good predictions of the sorption behaviour of divalent RNs, lanthanides and actinides on various clay rocks [7,8,17,18]. The difficulty for BC, and especially for the elements controlled by DOM, is being able to accurately predict their sorption behaviour, taking into account, on the one hand, the complex aqueous chemistry and, on the other hand, the complex mechanisms occurring at the solution–solid interface, both resulting from the relatively high carbonate and DOM concentrations. The work of Durce et al. [3] allowed the estimation of the complexation of BC DOM-Sn(IV) in conditions relevant for BC, but the transfer of the binary system to the ternary systems involving IdP or BC was unsuccessful without taking into account additive mechanisms, such as the formation of ternary complexes DOM-Sn(IV)-clay, or drastically adjusting the DOM-Sn(IV) interaction constant. DOM sorption on clay and the possible formation of ternary complexes RN-DOM-clay or DOM-RN-Clay was already demonstrated by several authors for different substrates and metals/RNs [20,45–47] though the extent and exact nature of the involved mechanisms was quite variable and uncertain. The interaction of DOM with clay and/or oxides is a complicated process, which often leads to sorptive fractionation and affects the ligand properties of DOM [46,48–50]. Due to this complexity, and to the resulting uncertainty on the real DOM sorption mechanism and its impact on RNs/Metals sorption onto clays, modelling ternary systems Clay-DOM-metals/RNs has proven not to be easy and is rarely predictive, especially for elements with a challenging chemistry, such as tetravalent metals [46]. This is even more true for complex mineral assemblies which also contain a significant amount of solid or sorbed organic matter, such as Boom Clay. The use of proxy pure clay systems to calibrate the sorption/complexation models allows refining of the CAA by accounting for processes specific to the ternary system. It gave here access to a rough estimation of the Sn(IV) sorption, in certain cases sufficient for performance assessment calculations. Yet, reaching an accurate prediction of the sorption on the natural rock/sediment was shown here to be difficult and would require a more in-depth understanding of the DOM aqueous complexation behaviour and sorption properties, and more experimental studies should be dedicated to this issue. In the absence of a clear understanding, experimental determination of *log* $R_d$ values for Sn(IV) in relevant conditions remains the most reliable option for a safety assessment of a deep geological nuclear waste repository in BC.

## 6. Conclusions

The present work had two parallel objectives. The first objective was to gain new insights on the Sn(IV) sorption on pure clay minerals and BC, in conditions relevant in the context of a nuclear waste repository. Sorption experiments were performed on Illite du Puy in the presence and absence of carbonates and in the presence and absence of dissolved organic matter. A sorption isotherm was also acquired on Boom Clay in present-day BC conditions. The sorption experiments performed on Illite du Puy showed that Sn(IV) strongly sorbs on clay minerals over the full pH range. The data obtained in the presence of carbonates showed decreased Sn(IV) sorption and evidenced the Sn(IV)–carbonate complexation and extrapolation of a conditional complexation constant. The presence of BC DOM was shown to strongly affect the Sn(IV) sorption, confirming the strong complexation of Sn(IV) with BC DOM.

The second objective of this study was to test the applicability of the component additivity approach for Sn(IV) sorption onto Boom Clay. To this aim, the 2SPNE SC model was used and calibrated on the results obtained on IdP. The lack of validated thermodynamic data describing the Sn(IV)–carbonate complexation was overcome by using the constant extrapolated from the sorption experiments on IdP. The Sn(IV)–DOM complexation constant previously published [3] did not provide a good description of the effect of DOM on the Sn(IV) sorption on IdP. The constant had to be adapted or sorption of the formed Sn(IV)-DOM complexes had to be considered to properly describe the experimental data. The component additivity approach integrating the aqueous complexation with carbonate and BC DOM and the possible formation of DOM-Sn(IV)-clay surface complexes provided a rough estimation of the Sn(IV) sorption on BC, but it failed in accurately predicting it, highlighting the high sensitivity of the model to the Sn(IV)–DOM complexation.

The present work extends the previously limited set of experimental data available on Sn(IV) sorption onto clay minerals and clay-rich host rock and provides $log\ R_d$ values directly applicable in safety calculations for a potential repository in Boom Clay. The modelling of these data revealed the need for a more mechanistic understanding and modelling of the role of OM on RN sorption on clay. This is especially needed in an OM-rich environment such as Boom Clay, and for radionuclides, for which the transport behaviour is highly linked to their complexation to DOM.

**Supplementary Materials:** The following supporting information can be downloaded at: https://www.mdpi.com/article/10.3390/min12091078/s1, Figure S1: Molecular Weight (MW) Distribution of DOM_NaClO$_4$; Figure S2: Eh-pH diagram for Sn; Figure S3: Eh-pH diagram for Sn; Figure S4–S9: variation of optimized constants with input parameter uncertainties; Table S1: Cationic composition of the clay suspensions; Table S2: Composition of the SPRING water.

**Author Contributions:** D.D. (conceptualization, methodology, validation, formal analysis, data curation, writing—original draft preparation,); S.S. (methodology, writing—review and editing); L.V.L. (writing—review and editing); L.W. (methodology, writing—review and editing); N.M. (writing—review and editing, project administration); S.B. (project administration). All authors have read and agreed to the published version of the manuscript.

**Funding:** The work presented herein has been performed in the broader framework of a public-public cooperation between ONDRAF/NIRAS and SCK CEN.

**Data Availability Statement:** The experimental data supporting this study can be found in Supplementary Materials.

**Acknowledgments:** We acknowledge D. Verhaegen for the experimental work.

**Conflicts of Interest:** The authors declare no conflict of interest. The funders helped in the design of the study and in the decision to publish the results.

**Appendix A**

The fraction of Sn lost during the centrifugation step is calculated such as:

$$Sn_{lost}(centrifugation) = \frac{Sn_0 - Sn_{eq}}{Sn_0}$$

with $Sn_0$ (mol/L), the concentration in the suspension before centrifugation and $Sn_{eq}$ (mol/L), the concentration in the supernatant after centrifugation. Both concentrations are calculated based on the measured activities using Equation (1).

The fraction of Sn lost during the sorption step is calculated such as:

$$Sn_{lost}(tube\ sorption) = \frac{Sn_{ini} - Sn_0}{Sn_{ini}}$$

with $Sn_{ini}$ (mol/ L), the concentration in the suspension at the beginning of the experiment calculated from the activity of the $^{113}$Sn spike solution using Equation (1).

**Table A1.** Fraction of Sn(IV) lost in the blank solutions during the contact time or during the centrifugation step.

| Conditions | Equilibrium pH | $Sn_{lost}$ (Tube Sorption) | $Sn_{lost}$ (Centrifugation) |
|---|---|---|---|
| 0.017 M NaClO$_4$ | 8.46 | 0.31 | 0.06 |
| | 8.39 | 0.36 | 0.03 |
| | 8.48 | 0.34 | 0.04 |
| | 3.01 | 0.70 | 0.50 |
| | 6.96 | 0.91 | 0.62 |
| | 11.97 | 0.14 | 0.05 |
| 0.015 M NaHCO$_3$ | 8.51 | −0.06 | 0.02 |
| | 8.55 | −0.08 | 0.03 |
| | 8.55 | −0.07 | 0.01 |
| 0.017 M NaClO$_4$ + 4.5 ppmC DOM_NaClO$_4$ | 8.38 | 0.08 | 0.02 |
| | 8.36 | −0.01 | −0.01 |
| | 8.40 | −0.01 | 0.04 |
| SPRING | 8.44 | −0.11 | −0.15 |
| | 8.41 | −0.08 | 0.01 |
| | 8.38 | −0.11 | −0.01 |

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
