# Peer review of "Sn(IV) Sorption onto Illite and Boom Clay: Effect of Carbonate and Dissolved Organic Matter"

_minerals, doi:10.3390/min12091078_

Round 1

Reviewer 1 Report

See attached pdf file

Reviewer 2 Report

The manuscript minerals-1708619 presents an experimental study on Sn(IV) sorption on clayey media (IdP and BC) in absence or presence of complexing species (carbonates and DOM). The experimental data interesting, especially regarding BC-DOM effect. Still such data requires a meticulous interpretation and modelling before publication (e.g. choice of model hypothesis, choice and accuracy of fixed and adjusted parameters). Currently, the study displays a limited set of input data, making tricky the modification of "blind" sorption models. For that reason, I advise to reject the manuscript in its present form. Publication with biased or under-constrained parameters would be detrimental to such interesting data. Few additional experiments and work on modelling (sensitivity analysis, accuracy on input constants) would really strengthen the corresponding study and improve its impact. A few comments are proposed below in this sense.

The dataset displays only five sorption isotherms. The first two pH and concentration isotherms show large uncertainties. Authors report tracer retention in absence of clay which may lead to significant bias (posible increase of [Sn]0SUSP when adding clay). A few additional data around pH 7-8, with different conditions (m/V ratio or concentration) would assess data repeatability and confirm its reliability. It is to note that isotherms with complexing compounds do not display similar issues. Indeed, complexation may decreases free Sn(IV) activity, thus preventing precipitation and increasing sorption of traces. Hence, data may suffer from precipitation rather than sorption on “tubes”. If not, this could easily be assessed by retention kinetics or by acidification of blanks.

An estimation of Sn(IV) solubility as a function of pH would be useful.     
Eventual Eh-pH diagram would also be welcome in appendix. 

A “predictive” SC model (Bradbury and Baeyens, 2009) is chosen and overestimation of Sn(IV) sorption further discussed. This deserves a sensitivity analysis regarding SC constants and their variability. As an example, a value log K(SSOH2+) = 4.0 is arbitrarily taken, while values between 4.0 and 5.5 are reported. It would be interesting to assess the effect of such variability (for all relevant constants) before further adjustment, or even better to get complementary data validating the chosen parameters. Instead, bibliography could be supplemented with various 2SPNE SC/CE dataset and tetravalent analogues.

The sorption isotherms in presence of carbonates and DOM seem accurate. Still the difference with or without complexing species is small (0.25 log) and corresponding adjustment deserves discussion. What is the confidence on the proposed Sn carbonate complexation constant (regarding variability of other constants and data uncertainties)? Would other experiments be more suitable to assess such complexation constant in solution in the first place?  

The last isotherm displays sorption of Sn(IV) on BC. The main difference with previous results comes from the content of clay within the solid. The choice of additional fitting parameters, due to heterogeneity of BC DOM is also questionable regarding all uncertainties.

The discussion section is a little short. It should be detailed or even merge with results section.

Calculation of uncertainties should be detailed (equation or included in xls file).

Reviewer 3 Report

Dear authors, your manuscript may be of interest to the reader. In my opinion, there are many inaccuracies and errors in the Introduction. This needs to be corrected.
1. What is the peculiarity of the sorption of tin (VI) on clay materials? You wrote that there are single studies, then write under what conditions (pH, temperature and something else).
2. Line 35-37: In the manuscript, authors are researching radioactive tin (VI). What information does the oxidation of divalent tin provide?

3. Line 43-45: Don't write countries where has been studied sorption of tin. Only provide citations. It is impossible to understand the sentence, what means 2: 1 ???
4. Line 45-47: It's hard to understand.
5. Line 50-52: write specifically about pH function… In my opinion, Line 48-61 is very difficult to understand what is already known and what you plan to explore. In addition, the efficiency of sorption of cations on the surface of clay materials depends on the pH of the aqueous solution, as well as the content of the clay mineral in the natural material. Write about it.

6. Illites and smectites are groups of minerals, not specific types of minerals. A group of minerals, according to certain characteristics, are united into Illites or Smectites.
7. 63-65: what are the illites and smectites present in the studied rock? What is their content and how their content has changed in depth?
8. What does "...batch sorption experiments .." mean?
9. What is o Illite du Puy (IdP)? The authors did not write anything about him
10. 82-86: the purpose of the work should be at the end of the Introduction!

In my opinion, the analytical information in the Introduction needs a very serious refinement + the authors need to understand what is Smectite, Illite. In the analysis of sorption, terms such as sorption capacity are used. In the introduction, the authors did not provide already known information about the sorption capacity of tin on montmorillonite (the authors wrote something about it).
In addition, the introduction should be shortened and leave the really necessary information! The design does not meet the requirements of the publisher.
11. What was NaClO4 used for? At what pH values ​​was the study conducted? 152: why do the authors believe that illite will dissolve ??? If the pH is very low, how can KCl help to it?
12. If the authors use the Langmuir model, it should be described in the Methodology section.
Dear authors, in my opinion, unfortunately, your manuscript contains a lot of errors. This applies to the structure of the article, the section Methodology is poorly related to the section results.
Conclusions need to be reduced! It is necessary to leave only the received concrete results.
In the manuscript, there is no literature for the last 5 years.

Reviewer 4 Report

          Abstract, line 13: which pH range is actually applied for the pH edge experiments, and the 8.4 value belongs to the isotherms.

          Abstract, line 20: here and afterwards we have CA, CAA and even CAA approach ...

          Keywords: It does not give added value for any indexing / search terms when recycling words from the title also for the keywords, here additional / alternative terms should be taken instead.

          Line 55: What was the concentration range of [Sn] in Kedziorek et al. 2007?

          Line 65: What are the concentration ranges for [Sn-126] expected due to release from a nuclear waste repository?

          Line 81: Here, the question of representativeness (and coupled on it, the composition and properties) of DOM pops up for the 1st time. Such assumptions should not be only based on similar conditional constants, here a much more detailed analyses is required, even when it is clear that the DOM composition will vary strongly. But any data transfer from the investigated systems to other (real) settings is very dangerous if no details about DOM specifics are known. This should be dealt with in more detail here.

          Line 155: What is meant by “disturbance of illite”?

          Line 158: Here we have the 1st occurrence of a value with an error assigned. This is handled very inconsistently throughout the manuscript: such an error is sometimes missing at all (without giving a reason for it), sometimes it is given but with no indication whether it refers to 1, 2 or even 3 sigma, sometimes it is one sigma, sometimes it is 2 sigma (or sigma is substituted by SD). I strongly advice to unify it (ideally to two sigma linking to 95% confidence intervals) and explain that at the very beginning so that any further mentioning can be avoided of how many sigmas the authors refer to.

          Line 172: Just giving an upper limit is insufficient. How many orders of magnitude can this value be higher than the maximum of Sn-126 expected (see remark to line 65)?

          Line 176: SPRING pore water is mentioned here for the first time, the explanation comes five lines below only.

          Lie 182. What is the overall composition of the SPRING pore water? Should be included at least in the supplement.

          Line 187: The paper should stick to SI units and not use such outdated units as Ci.

          Line 200: It is not clear, for which experimental points the high activity value is valid and for which one the lower one, with the difference being that drastic of more than 10%?

          Line 210: Epsilon is not explained, and neither is the derivation of the uncertainty assigned to epsilon, C_0 and A_0.

          Line 213: After lines 94 and 141 this is already the 3rd explanation what IdP means.

          Line 236: The cationic composition is mentioned here rather as a side line. But if one looks at the respective table ES.1 it turns out that the concentration of highly charged / strongly sorbing metal cations such as Al, Fe, Cu or Zn is well above the applied Sn concentrations. Thus their completion for sorption sites can not be simply neglected as done in this study. Probably this is one of the major reasons why the CA approach was not really successful as major players in the overall system were missing.

          Line 240: The concentration of an element in solution in equilibrium with a solid phase only depends on the activity in the solid but never on the solid/liquid ratio!

          Table 1: The resolution of the XRD method used here should be stated to justify listing phases such as Ankerite with just 0.1 (with an error of the same size it is questionable anyway).

          Line 253: The composition of the buffer should be given here.

          Line 277: Has there been any post-mortem analysis of the actual DIC content in the solution after the end of the experiment to check for outgassing effects?

          Line 282: Has it been checked by some blind samples (or by referring to respective literature) whether or not TRIS is interacting with either the mineral surfaces, the DOM or the Sn itself? Ignoring such effects may drastically falsify the interpretation of the experimental findings.

          Table 2: Are the values for [Sn] presented here are already adjusted for the loss of Sn presumably on the tube walls?

          Line 316: The electronic supplement does not contain any information about the Boom Clay experiments, only for IdP! Moreover, these Excel tables do also not contain information about the carbonate contents (when appropriate). And some column headings are not self-explanatory, so a bit more characters there would be helpful.

          Lines 339ff: Which extended Debye-Hückel model do the authors refer to (I can only guess that they mean the Davies approach)? And such models do not “correct the ionic strengths” but rather correct the activity according to changes in ionic strengths, most often going to infinite dilution.

          Line 349: When cation exchange is not implemented at all: what is the sense in providing respective selectivity coefficients in Table A1?

          Line 351: Is this assumption of Sn(IV) only sorbing on strong sites backed up by any references or own experiments?

          Table 5: A distinction between strong and weak sorbing sites on DOM does not make any sense when the log K values are identical ....

          Line 390: What is a “side reaction”?

          Lines 409ff: These explanations are confusing, it is an abridged explanations of the numerical approaches used to minimize a sum of squares of deviations between experimental and modelled points. But which values actually were used to model is not explained: Kd, log Kd, [Sn]_sorbed, [Sn]_dissolved, %Sn_sorbed or something different??

          Line 417: See remark to line 236. The reviewer has strong doubts that this simplification is justified.

          Line 419: What is the sulfate concentration? No values could be found in table ES.1 or elsewhere.

          Line 422: The following text is a repetition of earlier passages, but written less clearly. Can be dropped.

          Lines 457ff: Are there measurements to indicate the IdP surface charge as f(pH), e.g.by zeta potential measurements?

          Lines 466ff: The authors can also compare to the hydrolysis model from Ekberg and Brown: Hydrolysis of Metal Ions, Wiley-VCH, Weinheim, 2016 (2 volumes).

          Line 496: Why should harsher (what does this mean) centrifugation conditions enhance Sn(IV) tube wall sorption? Can you give references for such an effect?

          Figure 1, left part: As discussed in the text, SnO2(am) precipitation may be a reason for the observed high retention at low pH. Why has this not been checked in the model fitting phase?

          Figure 1, right part: This should be made a separate figure, placed already in the modelling section, and contain also SnO2(am) if applicable (difficult to judge as [Sn] is not given for this graph). Concerning the model from Bradbury and Baeyens 2009a: What was their original pH range, i.e. is the graph showing already significant extrapolation beyond the area of parameter derivation? Applying to ALL figures: the figure captions are very lengthy and verbose, “longer” than the figures themselves, should be shortened to a decent level.

          Line 537: Reduced by two or by a factor of two?

          Figure 2. The solubility limit of SnO2 should be added as a vertical line, as should be done for the natural background of [Sn]. And also the pH should be given (maybe just as a small text box within the graphs to not blow up further the captions. The lower part does not contain much additional information, maybe try to move the green line to the upper part (and the blue line does not deviate from the red one anyway).

          Figure 3. Again, the solubility limit of SnO2 should be added as a vertical line, as should be done for the natural background of [Sn].

          Appendix B: Why was the tube wall sorption of Sn not directly measured after digestion with a strong acid?

Author Response

Dear reviewer,

We thank you for your time. We have adapted the paper taking into account your comments and we hopefully succeeded in answering your concerns. You can find  more detailed answers in the attached document.

Kind regards

Round 2

Reviewer 1 Report

I find the adjustment of the papers justified.

Nevertheless some slight problems remains that I will let associate editor verify,

the caption in Figure 2 the Ks**(Sn-D  ) should be Ks**(Sn-DOM). please verify and correct.

the caption of Figure 3 is still noting Sn(OH)3CO3- (line 677 and 679). Please Correct!

Line 843: Capitalise the first name of Sébastien Savoye

line 929: correct "w eigths" and "w ith"

Author Response

Dear reviewer,

We thank you for your time and we are happy to see that the adaptations that we made to the paper answered your concerns. We also have corrected the minor errors that you mentioned.

Kind regards,

Delphine Durce

Reviewer 2 Report

The revised manuscript 1708619 details an experimental dataset on Sn(IV) retention on clay minerals. Compared to the initial manuscript, few additional data is added on Sn sorption isotherm at high concentration (Fig.2). The main modifications concern modelling, with a modification of strong site capacity for illite, a major revision of hydrolysis constants of Sn(IV) and a new analogy considered for carbonate-Sn(IV) aqueous complexes. This leads to a reprocessing of adjusted surface complexation constants, for Sn(IV)-Clay, DOM-Clay and DOM-Sn-clay surface species. The authors carefully took into account reviewer’s comments and the overall study deserves publication. Still the current revision rather confirms my first review: the lack of accurate input data fail to constrain satisfactorily the model parameters. For that reason, I propose to reject the current manuscript.            
To tackle the previous issue, authors need either more reliable thermodynamic data (e.g. hydrolysis and carbonates complexation constants), or a larger retention data set including other physicochemical conditions. In absence of such data, only a real sensitivity analysis may highlight the limitations on the proposed parameters. Here are two point to illustrate this comment:         
i) The choice to decrease sorption site quantity by a factor 2 is not really legitimate, while many other parameters are arbitrarily fixed (e.g. a value log K = 4.0 was taken for protolysis constant but it varies between 4.0. and 5.5 in literature). Modifying SC parameters would be less questionable than site capacity.            
ii) The important variation of model parameters between the previous and the revised manuscript raises questions:      
Analogy with Th(IV): log K[Sn(OH)3CO3] = 13.1, replaced by Zr(IV): log K[Sn(CO3)4] = 55.8,
Strong sorption site quantity decreased from 2.10-3 mol/kg(Illite) to 1.10-3 mol/kg,
Log Ks   [SClay-Sn-DOM] = 43.3 becomes 50.0 in the revised version,
Log Kw [wClay-Sn-DOM] = 39.4 becomes 46.0.                  
Log K6 [Sn-DOM] = 38.5 ± 0.1 becomes 45.4,
Log K7 [SClay-Sn-DOM] = 41.9 ± 0.1 becomes 48.9 ± 0.1,
Log K8 [WClay-Sn-DOM] = 41.9 ± 0.1 becomes 48.9 ± 0.1.
In the current state, the reported uncertainties (±0.1) would be misleading, as they do not represent the main variability (over 7 orders of magnitude) due to input uncertainties.
Even if such interesting experimental data and conceptual model deserve publication, it would be beneficial to perform a literature review on key inout data and a corresponding sensitivity analysis. This will help pointing out limitations of the proposed model and ease further use by the scientific community.           

Minor comments on Tables:   
All values of data used in the model should be copied out, even if equal to previously published data (e.g. Table 3). Blanks should only be used when the corresponding reaction is not considered.  
Table 4. Could include additional information on litterature values.
The reference to a database “thermochimieV10” should be replaced by the original references for each corresponding values.

Author Response

Dear reviewer,

We thank you for your time. We have adapted the paper taking into account your comments and we hopefully succeeded in answering your concerns. You can find  more detailed answers in the attached document.

Kind regards,

Delphine Durce

Reviewer 3 Report

I agree such version of manuscript

Author Response

Dear reviewer,

We thank you for your time and we are happy to see that the adaptations we made to the paper answered your concerns.

Kind regards

Reviewer 4 Report

The updated version of the manuscript addresses al previous comments by the reviewer in a satisfactory way.

Author Response

We thank the reviewer for the time he dedicated to our paper and for all his/her valuable comments. We are glad that the adaptations we made answered his/her concerns.

Regards,

Delphine Durce

Round 3

Reviewer 2 Report

We thank the authors for their meticulous answers. As already mentioned the experimental dataset is sound and interesting. The use of the available SC models shows some strong limitations. Regarding the current revision of the manuscript, the validity of the main input data is now discussed: site availability, speciation in solution, surface speciation. These limitations should be highlighted as much as possible in the discussion and conclusion, as they point out the further studies required on this topic. Given the above improvements, I advise to accept the manuscript minerals-1708619 for publication.

Author Response

(The authors gave the same response as above.)
